# Heparanase Modulates Chromatin Accessibility

**DOI:** 10.3390/cells12060891

**Published:** 2023-03-14

**Authors:** Honglian Li, Hua Zhang, Amelie Wenz, Ziqi Kang, Helen Wang, Israel Vlodavsky, Xingqi Chen, Jinping Li

**Affiliations:** 1SciLifeLab Uppsala, The Biomedical Center, Department of Medical Biochemistry and Microbiology, University of Uppsala, 75237 Uppsala, Sweden; 2Department of Immunology, Genetic and Pathology, University of Uppsala, 75237 Uppsala, Sweden; 3Technion Integrated Cancer Center (TICC), Rappaport, Faculty of Medicine, Technion, Haifa 31096, Israel

**Keywords:** heparanase, ATAC-seq, epigenetics, histone

## Abstract

Heparanase is the sole endoglucuronidase that degrades heparan sulfate in the cell surface and extracellular matrix (ECM). Several studies have reported the localization of heparanase in the cell nucleus, but the functional role of the nuclear enzyme is still obscure. Subjecting mouse embryonic fibroblasts (MEFs) derived from heparanase knockout (Hpse-KO) mice and applying transposase-accessible chromatin with sequencing (ATAC-seq), we revealed that heparanase is involved in the regulation of chromatin accessibility. Integrating with genome-wide analysis of chromatin states revealed an overall low activity in the enhancer and promoter regions of Hpse-KO MEFs compared with wild-type (WT) MEFs. Western blot analysis of MEFs and tissues derived from Hpse-KO vs. WT mice confirmed reduced expression of H3K27ac (acetylated lysine at N-terminal position 27 of the histone H3 protein). Our results offer a mechanistic explanation for the well-documented attenuation of inflammatory responses and tumor growth in Hpse-KO mice.

## 1. Introduction

Heparanase, the sole mammalian heparan sulfate (HS)-degrading endoglycosidase, belongs to the wider class of enzymes known as ‘retaining glycosidases,’ which catalyze hydrolytic cleavage of glycosidic bonds with net retention of anomeric stereochemistry [1]. It employs a conserved ‘double displacement mechanism’ involving two key catalytic amino acid residues—a nucleophile (Glu343) and a general acid/base proton donor (Glu225)—and transient formation of a covalent enzyme–substrate intermediate during the catalytic cycle [1]. The heparin/HS binding domains (HBD1, HBD2) are situated close to the active site micro-pocket fold. The heparanase mRNA encodes a 65 kDa pro-enzyme that is cleaved by cathepsin L into 8 and 50 kDa subunits that non-covalently associate to form the active enzyme. Structurally, heparanase is composed of a TIM-barrel fold that contains the enzyme’s active site and a flexible C-terminus domain required for secretion and the signaling function of the protein.

Heparanase regulates multiple biological activities in cancers, e.g., enhancing tumor growth, angiogenesis, and metastasis [2,3,4]. Heparanase expression is found to be elevated in almost all cancers examined including various carcinomas, sarcomas, and hematological malignancies. Numerous clinical association studies have consistently demonstrated that upregulation of heparanase expression correlates with increased tumor size, tumor angiogenesis, enhanced metastasis, and poor prognosis. In contrast, knockdown of heparanase or treatments of tumor-bearing mice with heparanase-inhibiting compounds markedly attenuate tumor progression, further underscoring the potential of anti-heparanase therapy for multiple types of cancer. Heparanase-neutralizing monoclonal antibodies block myeloma and lymphoma tumor growth and dissemination, attributable to a combined effect on the tumor cells and/or cells of the tumor microenvironment [5]. In fact, much of the impact of heparanase on tumor progression is related to its function in mediating tumor–host crosstalk, priming the tumor microenvironment to better support tumor growth, metastasis, and chemoresistance [4]. Given their heterogeneity and versatility, heparan sulfate proteoglycans (HSPGs) serve as important functional components of the cell surface and ECM. Hence, cleavage of HS by heparanase not only contributes to disassembly of the ECM, thereby facilitating cancer metastasis, but also affects diverse physiological and pathological processes ranging from gene transcription to DNA damage [6]. The known repertoire of the physio-pathological activities of heparanase is expanding. It activates cells of the innate immune system, promotes the formation of exosomes and autophagosomes, and stimulates signal transduction pathways via enzymatic and non-enzymatic activities. These effects dynamically impact multiple regulatory pathways that together drive tumor survival, growth, dissemination, and drug resistance [2,4,7,8]. Collectively, the emerging premise is that heparanase expressed by tumor cells, innate immune cells, activated endothelial cells, and other cells of the tumor microenvironment is a master regulator of the aggressive phenotype of cancer, an important contributor to the poor outcome of cancer patients, and a prime target for therapy. Apart from its role in cancer outcomes described in cancer-related studies, heparanase has also been found to play a significant role in modulating inflammatory responses [9,10]. The enzyme appears to fulfill some normal functions associated, for example, with vesicular traffic, lysosomal-based secretion, stress response, and heparan sulfate turnover. Based on these findings, heparanase inhibitors are developed for treatment of cancers and inflammatory diseases. Heparin-like compounds that inhibit heparanase activity are being evaluated in clinical trials for various types of cancer. Heparanase-neutralizing monoclonal antibodies are being evaluated in pre-clinical studies, and heparanase-inhibiting small molecules have been developed based on the crystal structure of the heparanase protein.

A key avenue through which heparanase accomplishes its multiple effects on cells and tissues is by regulating the bioavailability of HS-bound growth factors, chemokines, and cytokines, priming the tissue microenvironment. HS resides on the cell surface and the ECM in the form of proteoglycans (HSPGs), where, among other activities, it modulates cytokine bioavailability and functions as a co-receptor [11]. Binding of cytokines/growth factors to their specific cell membrane receptors is promoted by HS, followed by endocytosis of the growth factor–HS–receptor complex into late endosomes and lysosomes where it is catabolized by endo- and exo-glycosidases [12]. In this way, heparanase mediates tumor–host crosstalk, and promotes basic cellular processes (i.e., exosome formation, autophagy, and immune responses) that together orchestrate tissue remodeling [13]. Several studies have demonstrated the localization of HS and HSPGs in the cell nucleus [14,15]. It is unknown whether HS and HSPGs are transported into the cell nucleus after internalization and/or directly from the Golgi after synthesis. Nevertheless, it was demonstrated that exogenously added HS is transported into the nucleus [16], suggesting that secreted HS can be internalized into the cell nucleus.

Likewise, heparanase has been found in the cell nucleus [17]. Nuclear heparanase is thought to fulfill functional roles, such as regulating chromatin remodeling [18], chromosome stability [19], and gene transcription [20], depending or not depending on heparanase’s enzymatic activity. Early on, heparanase was believed to function primarily as a metabolic enzyme involved in HS/heparin catabolism [21]. Later on, a number of non-enzymatic activities (i.e., signal transduction, tumorigenesis, and gene transcription) were identified and ascribed to the enzyme C-terminus domain [22] and/or nuclear localization. However, little is known about the role of heparanase and HS within the nucleus. Using myeloma cell lines, it was found that: (i) within the nucleus, heparanase is present in the soluble fraction, and it is also bound to insoluble chromatin; (ii) the presence of nuclear heparanase enhances acetylation of histone H3 and promotes an open chromatin conformation; (iii) heparanase binds the promoter region of syndecan-1, MMP9, and CCND1, three genes whose expression is upregulated by heparanase; and (iv) heparanase increases phosphorylation of PTEN, leading to enhanced PTEN stability and thereby diminishing its function as a tumor suppressor [18]. Examination of available gene expression databases reveals that myeloma patients with high heparanase expression exhibited enhanced expression of acetyltransferase complexes and signaling pathways associated with myeloma growth and progression [18]. Together, these findings indicate that nuclear heparanase plays an important role in tumorigenesis by promoting chromatin remodeling that opens its conformation, allowing access to promotors of genes that drive tumor progression.

To gain insight regarding the role of nuclear heparanase, we applied the ATAC-seq for genome-wide analysis of the transposase-accessible chromatin genome-wide [23] in mouse embryonic fibroblasts (MEFs) isolated from wild-type (WT) vs. heparanase knockout (Hpse-KO) [24] mice. The results show an overall low activity in the enhancer and promoter regions in Hpse-KO MEFs compared with WT MEFs. Significantly lower levels of H3K27ac were detected in Hpse-KO embryonic cells and organs in comparison with WT tissues.

## 2. Materials and Methods

*Preparation of mouse embryonic fibroblasts (MEFs):* Heparanase knockout (Hpse-KO) (C57BL/6 genetic background) mice were generated as previously described [24] and maintained by littermates breeding in the animal facility of the Biomedical Center, Uppsala University, Sweden. Wild-type (WT) C57BL/6 mice were used as control. Embryonic fibroblasts (MEFs) were isolated from embryos of WT and Hpse-KO mice at day 14.5, as described [25]. Purified cells were cultured in Dulbecco’s modified Eagle’s medium (DMEM) (Thermo Fisher, Waltham, MA, USA, 11995073) containing 10% FBS (Thermo Fisher, 10500064) and 100 U/mL penicillin–streptomycin (Thermo Fisher, 15070063) in a humidified atmosphere containing 5% CO_2_ at 37 °C. The experimental protocol was conducted according to local ethical regulations.

*Transposase-accessible chromatin using ATAC-seq:* ATAC-seq was performed as previously described [26]. Briefly, both Hpse-KO and WT cells (50,000) were applied per ATAC-seq reaction. The transposition reaction was carried out according to the ATAC-seq protocol [26]. After transposition, the DNA was purified with MinElute PCR Purification kit (Qiagen, Hilden, Germany, 28004) and eluted in 10 µL Qiagen EB elution buffer. Sequencing libraries were prepared as described in the original ATAC-seq protocol. Sequencing was performed on Illumina NovaSeq 6000, and at least 20 million paired-end sequencing reads were generated for each ATAC-seq library.

*Data processing:* ATAC-seq raw data were mapped to mm10 with Bowtie2. The reads with mapping quality over 30 were kept and duplication reads were removed with Picard. MACS2 was used to perform peak calling with -q 0.01 -nomodel -shift 0 parameters. The peaks that overlapped with ENCODE genome blacklist were discarded. The read counts within peaks were generated with bedtools and the genome tracks were visualized using vig IGV.

*Differential peak analysis and genomic annotation:* Differential peak analysis was performed with DEseq2 package, keeping the differential peak list with threshold values of |logFC| > 1, FDR < 0.01. Genomic annotation was performed with the ChIPseeker package in R with seven genomic features: 3′ UTR, 5′ UTR, exon, intergenic region, intron, TSS, and TTS. The ChIP-seq data for ChromHMM were downloaded from Gene Expression Omnibus with the number GSE90895 and only the wild-type samples were downloaded. In total, 15 states were identified: Genic Enhancer 1 (EnhG1), Strong Transcription (Tx), Bivlant Transcription (BivTx), Repressed Polycomb (ReprPC), Bivlant Enhancer (EnhBiv), Weak Enhancer (EnhWk), Weak Transcription (TxWk), Active enhancer 1 (EnhA1), Genic Enhancer 2 (EnhG2), Active enhancer 2 (EnhA2), Active TSS (TssA), Flanking TSS upstream (TssFlnkU), Quiescent/low (Quies), ZNF genes & repeats (ZNF/Rpts), and Heterochromatin (Het). Bedtools documentation was applied to perform overlapping and to assign chromatin states into peaks. KEGG analysis was conducted with clusterProfiler package in R.

*Super-enhancer calling and function characterization:* Super-enhancers were identified with ROSE software with H3K27ac ChIP-seq data (GSE90803). Bedtools were used to analyze overlapping peaks between super-enhancer regions and chromatin-accessible regions. Ontological enrichment was performed by g: Profiler and visualized by Cytoscape setting FDR range from 0 to 0.001. Homer mm10 database and the super-enhancer bed files were used to calculate motif occurrences with a cut-off of *p*-value < 0.01.

*Western blotting:* Tissues of embryos and adult organs were collected from mice and homogenized with Polytron in ice-cold lysis buffer containing 20 mM Tris pH 7.4, 150 mM NaCl, 1% NP-40, 0.5% Sodium Deoxycholate, 0.1% SDS, and protease inhibitors. Cells were lysed in the same buffer and incubated on ice for 30 min. After centrifugation, the supernatants were collected and protein concentration was determined using the BCA method (Thermo Fisher, 23227). The lysates of tissues (60 µg total protein) and cells (20 µg total protein) were heated at 98 °C for 5 min before SDS-PAGE, followed by transferring to a polyvinylidene fluoride (PVDF) membrane (Cytiva, Marlborough, MA, USA, 10600023). After blocking with 5% non-fat milk for 1 h, the membranes were probed sequentially with primary and secondary antibodies. Antibody dilutions were: anti-H3K27ac 1:2500 (Abcam, Cambridge, UK, ab4729), anti-H3K27me3 1:1000 (Abcam, ab6002), anti-H3K4me1 1:1000 (Abcam, ab8895), anti-H3K9me3 1:1000 (Abcam, ab8898), anti-H3 1:1000 (Abcam, ab1791), and HRP-conjugated goat anti-rabbit IgG 1:10000 (Thermo Fisher, 31460). Signals were visualized using chemiluminescent HRP substrate (Sigma, Saint Louis, MS, USA, WBKLS0500). Band intensity was quantified by the Image Lab software (Bio-Rad, Hercules, CA, USA).

### Statistical Analysis

Statistical significance was analyzed by the 2-tailed unpaired Student’s *t*-test. Values of *p* < 0.05 were considered significant.

## 3. Results

### 3.1. Suppressed Activity of Promoters in Hpse-KO MEFs

Using the primary MEFs derived from Hpse-KO and WT mice, we collected two sets of ATAC-seq data. Examination of the heparanase gene loci identified enriched peaks in the heparanase promoter region of WT cells, which were missing in Hpse-KO cells, confirming the null expression of heparanase in the Hpse-KO cells (Figure 1A). Detection of the Coq2 gene is shown as a reference (Figure 1A). The high quality of ATAC-seq was reflected by highly concordant enrichment of the sequence reads at the transcription start sites (TSS) (Figure 1B) and the fraction of reads in peaks (FRiP) (Figure 1C). Furthermore, heatmap analysis showed good reproducibility of ATAC-seq from both WT and Hpse-KO samples (Figure 1D).

### 3.2. Identification and Characterization of Significantly Differential Peaks

A total of 13,000 ATAC-seq peaks downregulated or upregulated were identified (Figure 2A,B). Representative different peak regions are displayed with the genome browser (Figure 2C). To understand global functional changes in Hpse-KO cells, we systematically annotated the genomic feature distribution of ATAC-seq peaks and differential peaks and found the distinctive accessible chromatin patterns between WT and Hpse-KO cells (Figure 2D). A dramatic downregulation of the genes in the promoter region of Hpse-KO cells indicates that expression of a large number of genes may be reduced. In contrast, significant upregulation was detected in the distal intergenic region (Figure 2D). As this region contains multiple enhancers or inhibitory elements and is important in the regulation of gene expression, detailed genomic annotation is required to reveal the functional elements in this region of the Hpse-KO cells. Using chromHMM to characterize detailed chromatin states, we found 15 states that are located at the TSS and Flanking TSS upstream region (Figure 2E). Notably, among the 15 states, there are six histone markers, of which all are H3 with different modifications.

The 15 chromatin states were used to re-annotate the ATAC-seq peaks and the significantly differential peaks to highlight differences in the distribution of the chromatin states (Figure 2F). In Hpse-KO cells, the quiescent (Quies) regions of chromosomes increased significantly and an increase at the weak enhancer (EnhWk) region was also observed in comparison with WT cells (Figure 2F). In contrast, active enhancers (EnhA1, EnhA2) and transcription start sites (TssA, TssFlnkU) regions were significantly reduced in the Hpse-KO vs. WT cells. KEGG enrichment analysis of peaks in the quiescent region revealed a significantly enriched pathway of neuroactive ligand–receptor interaction. In comparison, KEGG enrichment analysis of the peaks in the weak enhancer regions showed major enrichment of several signaling pathways, including PI3K-Akt, MAPK focal adhesion, and calcium signaling (Figure 2G). In contrast, the active enhancer (EhA) and transcription start site (TSS) regions were mostly downregulated in Hpse-KO cells. KEGG analysis detected enrichment of a number of pathways, including PI3K-Akt, MAPK, and Ras signaling. These pathways are also enriched in the weak enhancer regions in Hpse-KO vs. WT cells (Figure 2H).

### 3.3. Suppressed Super-Enhancer in Hpse-KO MEFs

Representative SE-associated genes are marked as rank ordering of super-enhancers (Figure 3A). Gene ontology enrichment (GO) analysis of the SE-associated genes is shown in Figure 3B. Ontological analysis of SE-associated genes revealed their relationship with genes involved in important biological processes, including several biosynthetic and metabolic pathways as well as protein phosphorylation. The heatmap shows noticeably reduced chromatin accessibility levels of super-enhancers in Hpse-KO cells by overlapping and calculating peak counts (Figure 3C). The lower chromatin accessibility in Hpse-KO vs. WT cells can be seen in the super-enhancer region (e.g., Slc11a1) (Figure 3D). The pattern of super-enhancer transcription factor binding sequence obtained by motif analysis is shown in Figure 3E.

### 3.4. Histone Modifications of H3 in Hpse-KO MEFs and Organs

To determine if the epigenetic analysis of H3 in the EhA and TSS regions of Hpse-KO samples is of relevance to biological functions, we examined histone modification of H3 by Western blot analysis of the MEF cells. Using several anti-H3K antibodies recognizing different epitopes, we found that the level of H3K27ac was significantly reduced, while the level of H3K27me3 was substantially elevated in the Hpse-KO compared with WT cells. There was no difference in the levels of H3K4me1 or H3K9me3 between the two cell types (Figure 4A). These results are largely in agreement with the epigenetic analysis results. To reveal whether the reduced level of H3K27ac has a biological impact in mice, we analyzed H3K27ac in tissues dissected from the Hpse-KO and WT mice. Examination of brain, lung, heart, kidney, spleen, and pancreas did not detect a difference between Hpse-KO and WT organs. Interestingly, the level of H3K27ac was lower in the eyes of Hpse-KO compared with WT mice (Figure 4B). Examination of embryos under different developmental stages revealed a significantly lower level of H3K27ac in whole Hpse-KO vs. WT embryos (Figure 5), possibly indicating a role of H3K27ac in Hpse-KO mice during embryonic development.

## 4. Discussion

The sole endoglucuronidase in mammalians, heparanase, was thought to play a vital role in HS catabolism; surprisingly, elimination of the heparanase gene in mice did not lead to accumulation of HS, and only resulted in averagely longer HS chains [24]. The single observed phenotypic defect in Hpse-KO mice was in the retinal pigment epithelium [27], which may be associated with a low level of H3K27ac in the eyes (Figure 4B). Obviously, it appears that heparanase is not critical for HS catabolism and has no vital biological functions during embryonic development. The overall normal development of the Hpse-KO mice seems unaffected by the lowered level of H3K27ac detected in embryos (Figure 5).

Importantly, heparanase also displays non-enzymatic biological actions (e.g., cytokine production, signal transduction) [28]. Several observed impacts of heparanase expression on cellular signaling may or may not depend on the enzymatic activity [29,30]. The observed nuclear localization of heparanase further emphasizes the potential of heparanase’s non-enzymatic functions apart from its HS-degrading activity. Several studies have reported regulatory activities of nuclear heparanase, such as the induction of mammary cancer cell differentiation [31], regulation of glucose metabolism in endothelial cells [32], and gene transcription [33]. Some of these functions are attributed to the modulatory role of heparanase in chromatin packaging and remodeling [18], collectively, suggesting a stimulatory effect of nuclear heparanase on gene expression.

To gain some insight into the epigenetic regulation of heparanase function, we applied the advanced technique of ATAC-seq (assay for transposase-accessible chromatin using sequencing) to analyze the local accessibility of chromatin in embryonic fibroblasts (MEFs) derived from Hpse-KO and WT mice. Since the regions of gene transcription are controlled by cis-acting DNA elements, including enhancers, silencers, and promoters, we compared gene expression profiles in these regions. Super-enhancer is a special region in mammalian genomes driving transcription of important genes, controlling and defining cell identity. The EhA and TSS regions are chromatin states enriched with H3K27ac, the most frequently used marker of super-enhancer recognition [34]. Genes marked with this broad epigenetic domain, including H3K27 modification, are involved in cell identity and essential functions with strong clinical relevance [35]. The observed decreased level of H3K27ac in Hpae-KO MEFs and embryos is in line with previous reports showing that H3 methylation at actively transcribed genes is affected by heparanase [20] and that histone acetyltransferase (HAT) activity is modulated by heparanase through degradation of syndecan-1 [36]. Increased HAT activity in Hpse-high myeloma cells resulted in upregulation of transcription of multiple genes (i.e., VEGF, HGF, MMP-9, RANKL) that drive an aggressive tumor phenotype [36]. A novel set of genes under heparanase regulation has been characterized in T cells [20]. In this context, nuclear heparanase was shown to regulate the transcription of a cohort of inducible immune response genes by controlling histone H3 methylation, further expanding the transcriptional potential of heparanase. Heparanase was also shown to interact with promoters of multiple genes and micro-RNAs that upregulate gene transcription and control T cell differentiation. During herpes simplex virus-1 infection of corneal epithelial cells, heparanase translocates to the nucleus and enhances cytokine production [37].

The combined involvement of heparanase in epigenetic gene regulation and tumor progression has been further elucidated in studies demonstrating that chemotherapy, in addition to its cytotoxic effects on tumor cells, can support tumor re-growth and spread [38,39,40]. The possible involvement of heparanase in this observation is supported by our previous studies showing that heparanase expression is increased substantially in myeloma patients and cells treated with chemotherapy [40], providing a strong rationale for applying anti-heparanase therapy in combination with conventional anti-cancer drugs. In a subsequent study, we found that macrophages, an important constituent of the tumor microenvironment, are activated efficiently by chemotherapy (i.e., paclitaxel, cisplatin) and thereby support tumor growth. Strikingly, cytokine induction by chemotherapy was not observed in macrophages isolated from Hpse-KO mice [41]. Mechanistically, we found that chemotherapy (paclitaxel) stimulates the methylation of histone H3 on lysine 4 (H3K4) in wild-type but not Hpse-KO macrophages, leading to cytokine induction, and involving WDR5 [41]. This result provides another example of the involvement of heparanase in epigenetic gene regulation through histone modification.

Collectively, these results suggest a regulatory role of heparanase in chromatin packaging, accessibility, and activity. The increased activity of histone acetyltransferases (HAT) in cells expressing high levels of heparanase and the suppressed expression of H3K27ac and H3K4 tri-methylation (histone methyltransferases) in Hpse-KO cells/tissues suggest an important regulatory function of heparanase in histone modification and activity. The overall low activity in regions that control gene transcription in Hpse-KO MEFs may be associated with suppressed inflammatory responses and tumor growth in Hpse-KO mice [10,42,43]. Indeed, overexpression of heparanase in mice promotes tumor initiation and growth [44,45] and stimulates inflammatory responses [7,9,46]. Collectively, we propose that heparanase may function as an epigenetic marker, modulating innate immunity and other cellular functions, in part by controlling gene expression. Given that heparanase is a well-recognized target for inhibition of cancer progression and inflammation [43,47], our findings further strengthen this notion by broadening the multitasking molecular function of the heparanase gene and protein.

## Figures and Tables

**Figure 1 cells-12-00891-f001:**
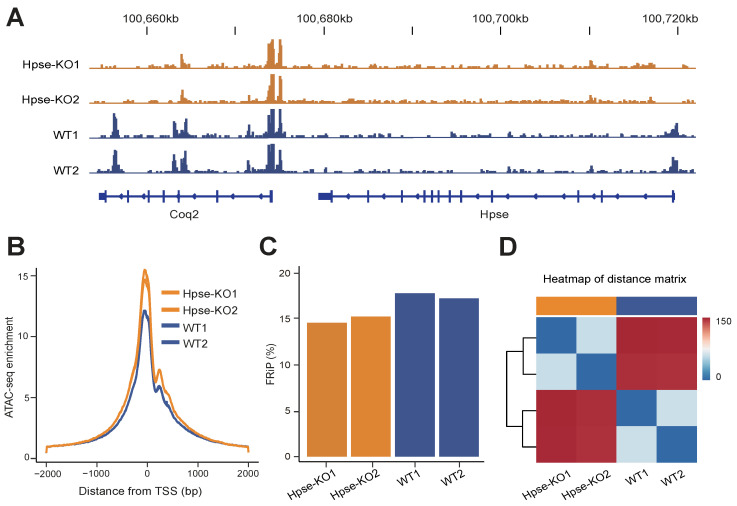
*Quality of chromatin accessibility datasets.* (**A**) Representative genome browser tracks of ATAC-seq data for heparanase and Coq2 genes. (**B**) Enrichment results of ATAC-seq reads at TSS. (**C**) Fraction of reads in peaks (FRiP) for different samples. (**D**) Heatmap of distance matrix between WT and Hpse-KO samples.

**Figure 2 cells-12-00891-f002:**
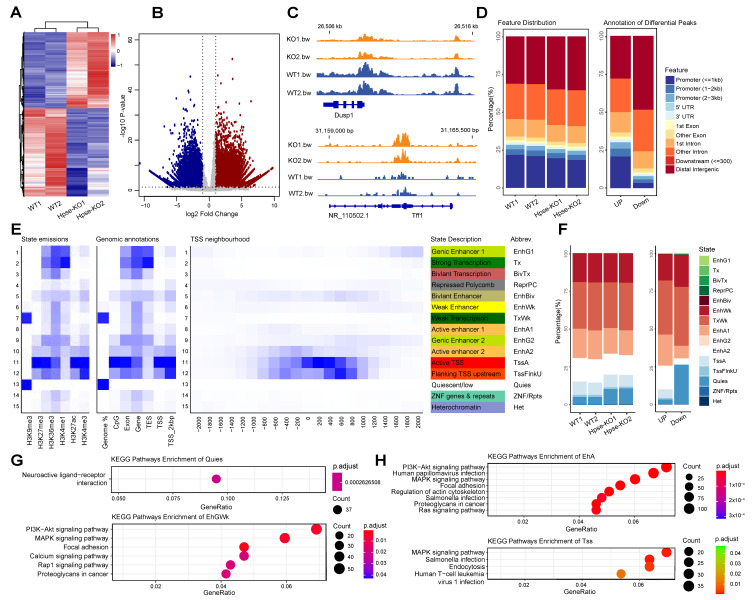
*Identification and characterization of differential peaks.* (**A**) Heatmap of normalized peak counts. (**B**) Volcano plots of differential peaks identified; downregulated peaks are indicated in blue and upregulated peaks are indicated in red. (**C**) Representative genome browser tracks of differential peaks in Dusp1 and Tff1 regions. (**D**) Genomic feature distribution of ATAC-seq peaks. (**E**) ChromHMM defined chromatin states with six histone markers. (**F**) Bar graphs, obtained based on the chromHMM results, showing the chromatin states distribution of ATAC-seq peaks. (**G**) Kyoto Encyclopedia Genome (KEGG) analysis showing significantly upregulated regions (Quies and EhGWk) in Hpse-KO samples. (**H**) KEGG analysis showing significantly downregulated regions (EnhA and TSS) in Hpse-KO samples.

**Figure 3 cells-12-00891-f003:**
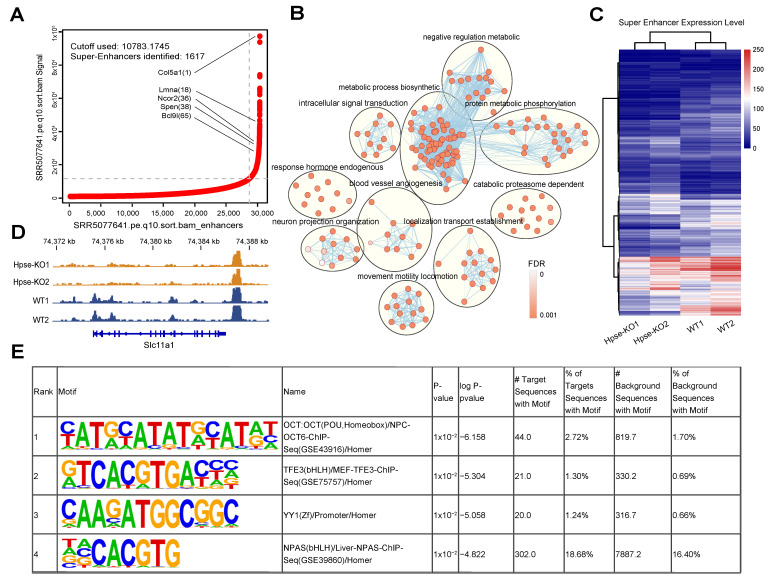
*Identification and functional characterization of super-enhancers.* (**A**) Rank ordering of super-enhancers with representative SE-associated genes. (**B**) Ontological enrichment of SE-associated genes. (**C**) Heatmap displays the peak counts of super-enhancers in WT and Hpse-KO cells. (**D**) Genome track showing reduction of ATAC-seq signal in the super-enhancers region of Hpse-KO cells. (**E**) Motif analysis of super-enhancers.

**Figure 4 cells-12-00891-f004:**
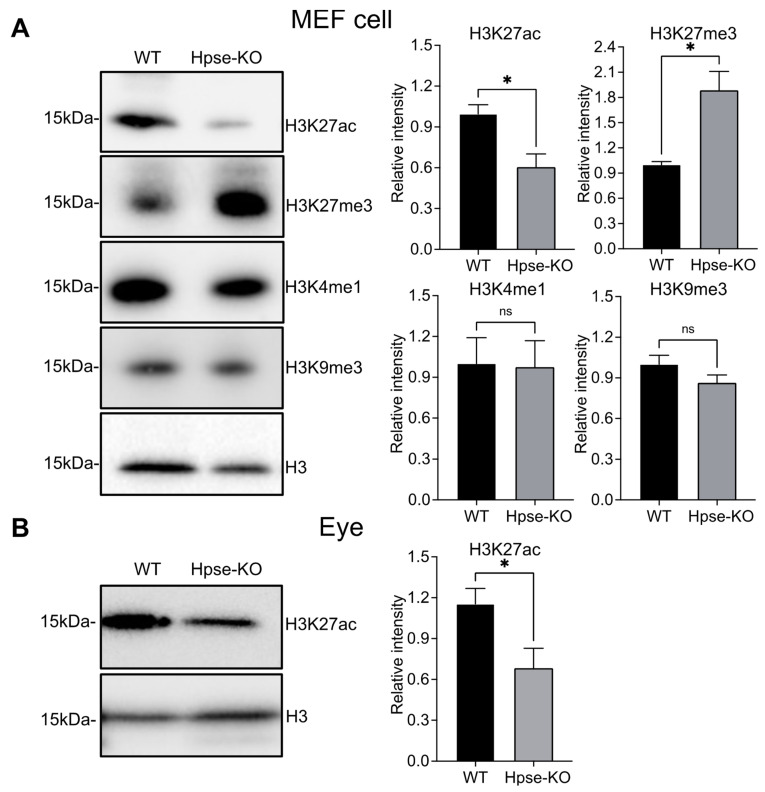
*Assessment of H3K levels in cells and organs by Western blot analysis.* (**A**) Cell lysates of 20 µg total protein were separated by SDS-PAGE and probed with specific antibodies against H3K27ac, H3K27me3, H3K4me1, and H3K9me3. Intensity of the bands was quantified by the Image Lab software (Bio-Rad). H3 is used as loading control. (**B**) Protein levels of H3K27ac in the eyes of adult mice analyzed by the same method using anti-H3K27ac antibody. A representative blot is shown. The graphs show the mean ± SEM of band intensity (WT is set as 1). Relative expression related to WT is shown *: *p* < 0.05; ns = no significant difference. The data were from three independent cultures; the samples of eyes were collected from 6 mice per group.

**Figure 5 cells-12-00891-f005:**
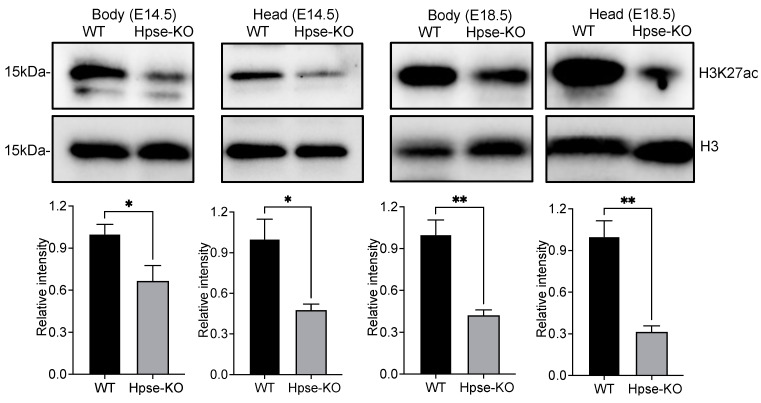
*Western blot analysis of H3K27ac in embryos.* Upper panels show representative blot from three experiments; lower panel shows relative intensity between WT and Hpse-KO. H3 represents internal control. The results are shown as the mean ± SEM of the band intensity. *: *p* < 0.05; **: *p* < 0.01.

## Data Availability

Not applicable.

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
