# Peer review of "Heparanase Modulates Chromatin Accessibility"

_cells, 2023, doi:10.3390/cells12060891_

Round 1

Reviewer 1 Report

The manuscript by Li et al. represents a comparative analysis of transposase-accessible chromatin with sequencing (ATAC-Seq) data for mouse embryonic fibroblasts (MEFs) derived from wild or heparanase knockout (Hpse-KO) mice. Genome-wide analysis of chromatin states revealed an overall low activity in the enhancer and promoter regions of Hpse-KO MEFs compared with wild-type (WT) MEFs, which allowed the authors to conclude about an involvement of heparanase in modulation of chromatin accessibility. In general, this is an original and very interesting study in a scientific field that still remains little studied. The manuscript is well written and bioinformatics analysis of the obtained results is presented in details, which was complemented with Western blot data for different histone modifications of H3.

However, one point will benefit from additional explanation.

1. The Figure 6 is described by the only phrase (“Assessment of cell proliferation supports a restrained proliferation rate of Hpse-KO vs WT MEFs (Fig. 6.)”). It would be suitable to explain why the MTT test (which mainly indicates the viability of cells, and not their proliferative activity) was used in this study, and why only three time points were accessed.

2. It would be good to check the manuscript again for misprints.

Author Response

Response to Reviewer 1 Comments

Point 1. The Figure 6 is described by the only phrase (“Assessment of cell proliferation supports a restrained proliferation rate of Hpse-KO vs WT MEFs (Fig. 6.)”). It would be suitable to explain why the MTT test (which mainly indicates the viability of cells, and not their proliferative activity) was used in this study, and why only three time points were accessed.

Response to point 1:

The proliferative activity of heparanase is controversial, indirect, cell type specific, and often marginal. The reviewer is right in stating that the MTT assay measures cell viability rather than proliferative capacity. Given these considerations, we decided to omit figure 6 altogether and address this issue in more detail (i.e., various cell types derived from heparanase knockout vs heparanase overexpressing mice, rescue experiments using recombinant enzyme, serum dependence, as well as in vivo studies using BrdU incorporation/Ki67). In these experiments we will apply the XTT assay and actual daily cell counting (instead of MTT) as indicated by the reviewer. Such a detailed study is beyond the scope of the present work and, in fact, appears unrelated to the main issue of heparanase and chromatin accessibility.

Point 2: It would be good to check the manuscript again for misprints.

Response to point 2:

Manuscript was rechecked for misprints/spelling mistakes.

Reviewer 2 Report

The authors in this paper are investigating the role of nuclear heparanase (HPSE) in regulating chromatin accessibility and therefore access to enhancer and promoter regions. They have used ATAC-Seq to measure HPSE-driven changes in chromatin accessibility in embryonic fibroblasts derived from WT vs HPSE KO-mice. The main findings reported in the paper include:

1.       Lower activity of enhancer and promoter regions of KO vs WT suggesting nuclear HPSE affects chromatin structure

2.       Reduced H3K27 acetylation in KO vs WT mice

3.   Reduced proliferation rates for KO vs WT fibroblast cells which were interpreted as being linked to lower activity of enhancer and promoter regions.  

Although the findings of HPSE and heparan sulfate in the nucleus is provocative and potentially highly important to our understanding of glycosaminoglycan biology, I have outstanding questions to be addressed before recommending the manuscript for publication:

·    The paper does not definitively establish the functional role(s) of HPSE in cell proliferation. As acknowledged by the authors, HPSE affects the extracellular matrix by trimming heparan sulfate and thereby altering the presentation of growth factors and chemokines. Therefore, the authors must resolve the role HPSE plays in the nucleus vs extracellular matrix/cell surface as it relates to cell proliferation.

·         Does the addition of extracellular HPSE rescue the KO cell growth vs WT? 

Author Response

Response to Reviewer 2 Comments

Point 1. The paper does not definitively establish the functional role(s) of HPSE in cell proliferation. As acknowledged by the authors, HPSE affects the extracellular matrix by trimming heparan sulfate and thereby altering the presentation of growth factors and chemokines. Therefore, the authors must resolve the role HPSE plays in the nucleus vs extracellular matrix/cell surface as it relates to cell proliferation.

Response to point 1

We thank the reviewer and fully agree with the statement that the role that HPSE plays in the nucleus vs extracellular matrix/cell surface as it relates to cell proliferation (and other functions), has not been resolved. That was not our intention in the first place and, in fact, appears unrelated to the main issue of heparanase and chromatin accessibility. While the release of HS-bound growth promoting factors by extracellular heparanase is straightforward, the contribution of nuclear heparanase and even the mode by which heparanase enters the nucleus are far from being resolved.

Notably, the function of heparanase in cell proliferation is controversial, indirect, cell type dependent, and often marginal. Given these considerations, we decided to omit figure 6 altogether and address this issue in more detail (i.e., various cell types derived from heparanase knockout vs heparanase overexpressing mice, rescue experiments using recombinant enzyme, serum dependence, as well as in vivo studies using BrdU incorporation/Ki67). In these experiments we will apply the XTT assay and actual daily cell counting (instead of MTT). Such a detailed study is beyond the scope of the present work.

Point 2. Does the addition of extracellular HPSE rescue the KO cell growth vs WT? 

Response to point 2

The proposed 'rescue' experiment is fundamental and, as detailed above, will be perfrmed in a separate study focusing on the functions of extracellular vs nuclear heparanase. Hence, figure 6 and the entire of issue of heparanase and cell proliferation, are no longer addressed in the present study.

Round 2

Reviewer 2 Report

The authors have addressed my major concerns